# How to promote the quality of technical education? A fuzzy-set qualitative analysis of 16 technical colleges in China

Fan Zhang[1,2¤], Yile Liu[3], Lianhua Fan[4]*

1 School of Education, Shaanxi Normal University, Xi'an, China, 2 E-commerce Department, Guangdong Electronic-Commerce Technician College, Guangdong, China, 3 Center for Teacher Education Research, Beijing Normal University, Beijing, China, 4 School of Education, Xi'an International Studies University, Xi'an, Shaanxi Province, China

¤ Current Address: E-commerce Department, Guangdong Electronic-Commerce Technician College, Guangdong, China

* 834733932@qq.com

## Abstract

This study focuses on how technical and vocational education and training (TVET) institutions can effectively enhance educational quality to cultivate a globally competitive industrial workforce, addressing the World Banks projection that over 1.1 billion jobs worldwide will undergo skill transformations in the next decade due to automation, digitalization, and the green economy transition. Employing the fuzzy-set qualitative comparative analysis (fsQCA) method, the research conducts an in-depth analysis of developmental data from 16 Chinese TVET institutions to identify key factors influencing their competitiveness. The findings reveal that social service constitutes the foundation of high-quality vocational education. The social service capacity of TVET institutions is primarily reflected in vocational skill training, aligning with the core philosophy of Singapore Polytechnic's "SkillsFuture" initiative. Through data analysis, a "social service-driven" development mechanism is identified: under similar conditions, TVET institutions achieve high-quality development by participating in government-funded vocational training programs. Simultaneously, two types of developmental bottlenecks are uncovered: (1) the "student skill level-international exchange constraints" type, where limited student proficiency and international collaboration hinder institutional progress; and (2) the "social service-technological R&D constraints" type, where weak social service delivery and technology transfer capabilities act as critical barriers. The outcomes provide a robust reference for global TVET stakeholders and policymakers to optimize industrial talent cultivation strategies and deepen integration into global value chains. The methodological framework also holds transferability to other domains, enabling regions to pinpoint success pathways and avoid ineffective measures.

**Data availability statement:** All relevant data are within the paper and its Supporting Information files.

**Funding:** the 2024 Guangdong Provincial Research Topic on Technical Education and Vocational Training: A Study on the Compatibility of Talent Cultivation Models in Guangdong Technical Colleges with Industrial Structure Upgrading and Transformation (Award No. KT2024052 to Fan Zhang); (2) the 2024 National University and Vocational College Logistics Teaching Reform and Research Project: Research on Teaching Reform and Training Mode of Cross-border E-commerce and Cross-border Logistics Compound Talents in Technical Colleges Guided by OBE Concept (Award No. JZW2024262 to Fan Zhang); and (3) the 2023 Western Project of the National Social Science Fund of China in Education: Research on the Supervision and Evaluation Mechanism of Digital Empowerment for Quality and Balanced Development of Compulsory Education (Award No. CHX230350 to Lianhua Fan). The funders had no role in study design, data collection and analysis, decision to publish, or preparation of the manuscript.

**Competing interests:** The authors declare no financial/personal relationships that could be construed as potential conflicts.

## Introduction

The role of technical and vocational education and training (TVET) in addressing industrial demand for high-quality technical talents has been widely discussed in existing literature (Zhou & Jing, 2021; Yang & Dong, 2012) [1,2].As a core pillar of global education reform, TVET quality improvement is not only tied to regional industrial upgrading but also characterized by its distinctive social-service function—TVET providers offer publicly accessible vocational-upgrading courses that feed directly back into curriculum innovation, creating a dynamic loop between skill training and educational optimization. Benchmark systems worldwide have validated the value of this function: Singapore Polytechnic's "SkillsFuture" initiative [3]integrates industry needs into skill development, Germany's dual system [4] combines classroom learning with on-the-job training, and Australia's industry-led standardsalign vocational programs with labor market demands [5]—all of which illustrate how practice-oriented programmes and employer-engaged governance can expand human-capital dividends and raise employment rates [6–8].Globally,government-funded vocational training programs have further scaled this impact, such as the World Bank-supported TVET project in Mozambique, which links transportation infrastructure and skill training to boost regional employment [9].

For China, a key node in the global industrial chain, technical colleges (i.e., TVET institutions) have continuously absorbed such advanced international experience to advance educational reform [10]. Yet they still face three persistent challenges that hinder high-quality development: first, low social recognition of technical professions, which undermines talent attraction and retention [11,12]; second, uneven regional resource distribution, despite efforts to mitigate gaps through alliance-based education and training base optimization [13–16]; and third, fragmented responsibilities among schools, enterprises, and the government, a issue that also plagues global TVET systems and weakens collaborative efficiency [17,18]. These challenges are compounded by China's dual pressure: emerging sectors like smart manufacturing and the digital economy face pronounced skill shortages, while traditional industries demand reskilling for incumbent workers—making the need to address TVET development bottlenecks even more urgent [19].

Previous studies on TVET quality have mostly relied on macro policy analysis or single-case illustrations, with some exploring isolated factors such as public training base sustainability or enterprise-school cooperation mechanisms. However, these studies have two critical limitations: they rarely empirically validate how strategies adapt to China's diverse regional contexts, and more importantly, they overlook the combined effects of multiple strategies—failing to capture how different condition combinations may equally drive high-quality development. This gap stems from the limitations of traditional analytical methods, which struggle to unpack the complex, non-linear relationships between TVET's multi-dimensional influencing factors.

Fuzzy-set Qualitative Comparative Analysis (fsQCA) is well-suited to address this limitation, as it enables us to not only identify equifinal pathways for TVET high-quality development (given the complex, multi-condition interactive nature of TVET

development, where different condition combinations can lead to the same high-quality outcome) but also summarize typical bottleneck types through the analysis of condition necessity and sufficiency [20]. Against this backdrop, the present study builds upon quality cultivation theory while incorporating distinctive features of vocational training to propose an influential factors model for technical colleges' quality development. Taking 16 Chinese technical colleges as research objects, this study utilizes administrative data sources—including institutional annual quality reports and local government TVET development bulletins—to empirically examine these factors. The paper proceeds systematically: (1) synthesizing theoretical foundations and existing TVET quality improvement literature to delineate research gaps; (2) detailing the research design, encompassing case selection protocols, data sources, variable operationalization, and fsQCA methodology; (3) presenting fsQCA results that reveal key configuration pathways and bottleneck constraints in quality development; (4) discussing theoretical contributions and practical implications aligned with global TVET development needs; and (5) concluding with study limitations and future research directions.

## Literature review

### 1. Evolving dimensions of TVET quality assessment

Scholarly consensus identifies TVET quality as a multi-faceted construct, commonly assessed through three interconnected dimensions: educational processes, outcome outputs, and institutional/systemic guarantees [21,22].

Educational Processes.Key foci include the adaptability of teaching resources to industry needs [23]and the critical importance of teacher professional development, particularly workplace learning integrating industry practice [24,25]. The effectiveness of pedagogical approaches is paramount.Structured coaching and mentoring frameworks, grounded in General Systems Theory, enable sustainable development of pedagogical and professional competencies for TVET lecturers, bridging skills gaps and aligning educator capabilities with dynamic labor market demands [26].

Outcome Outputs. Assessment extends beyond traditional academic metrics to encompass graduate employment quality, entrepreneurship outcomes, and crucially, the social service contribution of institutions [7,27] This includes vocational training for the public, skills certification, and industry-specific upskilling programs, recognized as vital for alleviating structural employment imbalances and demonstrating direct socio-economic impact [22]. Achieving precision in aligning curricula with evolving industry demands is consistently highlighted as a critical success factor [28].

Institutional/Systemic Guarantees. This dimension examines the enablers of quality, covering policy effectiveness and funding mechanisms [29], infrastructure quality (training facilities), industry-academia collaboration depth, international engagement levels, and crucially, models for resource collaboration (e.g., alliances, shared training bases) and multi-stakeholder governance involving government, industry, and educational institutions [30].

Scholars further emphasize that reform should be closely combined with enterprise research and learningto achieve deep collaboration between education and industry [31]. However, contemporary challenges like digitalization, the green transition, and the demand for 'future skills' underscore a limitation: existing dimensional frameworks often struggle to systematically incorporate these rapidly evolving priorities, potentially overlooking key drivers of modern TVET relevance and quality [32].

### 2. Evaluation models for high-quality technical vocational education

In terms of evaluation models, the CIPP model and Triple Helix theory provide important analytical frameworks for quality research in technical vocational education. The CIPP model consists of four evaluation stages: context, input, process, and output [33]. Based on this, Yu Minghui (2015) proposed five key indicators: "student quality, teachers, hardware, school-enterprise cooperation, and service to the economy and society [34]," which match the evaluation elements of technical vocational education level and output. Zong Cheng et al. proposed the dimension of international cooperation quality in the *Annual Report on Vocational Education Quality* [35], and Cai Wenbo constructed seven dimensions,



including "student skill level, faculty professional development, technical R&D capability, infrastructure construction, industry-university-research integration, international exchange and cooperation, and social service contribution" [5], comprehensively reflecting the contribution of technical vocational education to social development. The Triple Helix theory proposed by Etzkowitz and Leydesdorff (2000) reveals the influence mechanism of interaction among the government, industry, and colleges on educational quality from the perspective of collaborative innovation [36]

These two models have their respective advantages: the CIPP model is more suitable for static and structured quality assessment, while the Triple Helix theory is better at analyzing dynamic and systematic collaborative innovation processes [37]. Existing studies have not fully integrated the advantages of these two models, and it is advisable to consider constructing a composite model that integrates static assessment and dynamic analysis to more comprehensively grasp the development laws of technical vocational education quality.

### 3. Literature on fsQCA method in research on high-quality improvement paths

The fuzzy-set qualitative comparative analysis (fsQCA) method, which deconstructs configurational causality through Boolean algebraic logic [38], has proven its value in higher education research [39].and studies on industry-education integration policies [40]. However, its application in technical and vocational education faces adaptability constraints: 96% of research subjects in existing studies are derived from conventional universities [41]

The application of the fsQCA method in educational quality research has broken through the limitations of traditional quantitative research, providing new ideas and methods for revealing the complex causal mechanisms of educational quality improvement. The development level of technical vocational education is a key indicator for measuring the quality and efficiency of skilled talent cultivation in a region, which is crucial for matching the supply of skilled talent with the needs of industrial upgrading. The application of the fsQCA method in technical vocational education quality research shows unique value. However, existing applications still have obvious limitations: on the one hand, the sample selection focuses on general higher education institutions, insufficiently considering the particularity of technical vocational colleges; on the other hand, variable settings fail to fully reflect the impact of new trends such as digital transformation on technical vocational education. To apply this method in research on technical vocational colleges, it is necessary to increase investigations into regional technical vocational colleges, introduce emerging variables with technical vocational attributes, and attempt to combine fsQCA with case studies to gain a deeper understanding of the action mechanisms behind configurational effects.

## Background and hypothesis development

### Theoretical model

When constructing an evaluation model for the high-quality development of technical vocational education, this study further integrates the three dimensions of the Triple Helix theory (government, colleges, enterprises) [36]on the basis of the seven dimensions proposed by Cai Wenbo to construct a comprehensive evaluation model [7] (shown in Fig 1). The government dimension reflects policy support through funding vocational training projects; the college dimension covers student skill level, teacher professional competence, infrastructure construction, international exchange and cooperation, and research capability; the enterprise dimension reflects the actual level of school-enterprise cooperation through industry-education integration. See Table 1 for details.These elements interact to jointly promote the high-quality development of technical vocational education and improve educational quality and student training effectiveness.

In the evaluation model for the high-quality development of technical vocational education, seven variables are defined based on technical vocational attributes (see Table 2). The government dimension is reflected by social service contribution (GVTP), which mainly measures the participation of technical vocational colleges in social service contributions, including the number of trainees and the number of vocational qualification certificates obtained. The college dimension



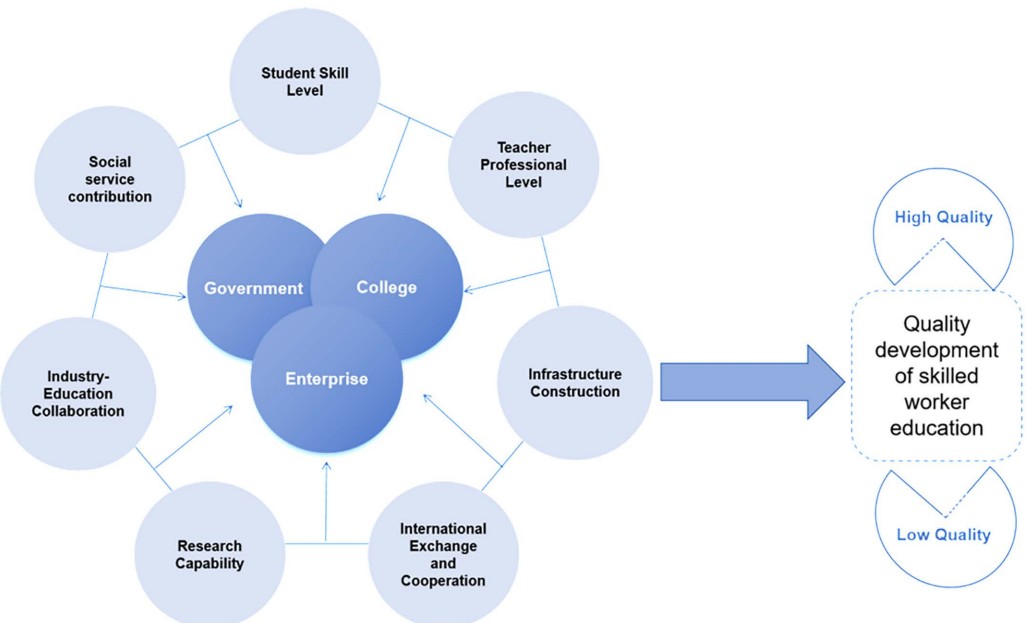

**Fig 1. Analysis framework of quality development of skilled worker education.**

**Table 1. Summary of the variables used in the model.**

| S/N | Dimension | Variable | Abbreviation | Definition |
|---|---|---|---|---|
| 1 | Government | Social service contribution | GVTP | It specifically refers to the vocational training projects funded by the government to improve social employment skills and in which technical colleges participate, mainly studying the number of people trained in the projects and the number of qualified vocational qualifications obtained. |
| 2 | College | Student Skill Level | SSL | Indicators measuring the skill level of students, including the total number of students and the number of awards received in provincial-level or above skill competitions. |
| | | Teacher Professional Competence | TPC | Indicators assessing teachers' professional status, encompassing the total number of teachers, the percentage with a graduate degree or above, the proportion of senior lecturers, honors received, and awards in provincial-level or above skill competitions. |
| | | Infrastructure Construction | IC | Indicators representing the institution's infrastructure, including the number of training rooms, revenue from fiscal appropriations in 2024, and average area per student. |
| | | International Exchange and Cooperation | ICE | Indicators showing the extent of international cooperation, covering geographic latitude of international exchange, dimensions of international communication types, and the number of cooperative education projects. |
| | | Research Capability | RC | Indicators reflecting the research output of the institution, such as the number of papers published per school, patent applications and grants for teachers and students, and projects approved by the school. |
| 3 | Enterprise | Integration of Industry and Education | IIE | Indicators gauging the depth and breadth of collaboration between the institution and industry, including the number of school-enterprise cooperation projects and the scale of cooperative enterprises. |

covers five aspects: student skill level (SSL), teacher professional competence (TPC), infrastructure construction (IC), international exchange and cooperation (ICE), and research capability (RC), evaluating educational quality from multiple perspectives such as students' competition awards, teachers' academic qualifications and honors, school hardware facilities, the breadth of international cooperation, and scientific research achievements. The enterprise dimension focuses

**Table 2. CIPP vs. triple helix frameworks in technical education research.**

| Model | Strengths | Limitations |
| --- | --- | --- |
| CIPP | Static, structured assessment | Limited dynamic analysis |
| Triple Helix | Systemic innovation processes | Lack of quantitative metrics |

on industry-education integration (IIE), measuring the degree of school-enterprise collaboration through the number of school-enterprise cooperation projects and the scale of cooperating enterprises. These variables together form a comprehensive evaluation system for technical vocational education quality, providing data support for in-depth analysis of the development of technical vocational education.

## Research method

### Qualitative comparative analysis (QCA) and Fuzzy-set QCA (fsQCA)

Rooted in case comparison principles and set-theoretic Boolean algebra, qualitative comparative analysis (QCA) is a robust methodology widely employed in qualitative research to test theoretical hypotheses and uncover complex causal configurations [42]. By examining combinations of factors, drivers, and antecedents, QCA identifies how these elements collectively give rise to specific outcomes [43], overcoming the limitations of traditional variance-based approaches that often overlook interactive effects [44]. This flexibility makes QCA particularly suitable for exploring the multifaceted mechanisms underlying technical education quality, as it reveals non-linear relationships and compensatory pathways among variables. Although QCA is suitable for small-sample research, excessively small sample sizes may restrict its ability to identify complex causal relationships. Small samples can lead to insufficient cases for certain condition combinations, compromising the comprehensiveness of the analysis and the reliability of conclusions. Additionally, the generalizability of findings from small-sample studies to broader contexts may be limited.

Fuzzy-set QCA (fsQCA), an advanced variant, specializes in analyzing "joint effects" and interactive relationships, thriving in small-sample research by transforming qualitative data into fuzzy-set numerical values (0–1) through logical calibration. This process preserves the richness of qualitative insights while enabling quantitative comparison, thus bridging the gap between depth and generalizability. fsQCA requires transforming data into fuzzy-set membership scores, a process involving researchers' subjective judgments. Determining calibration parameters (such as threshold values for full membership and non-membership) can vary across researchers, affecting the objectivity and comparability of analysis results. This subjectivity may invite research findings, particularly when clear calibration standards are absent.

Despite these limitations, fsQCA is well-suited for studying educational quality issues. Educational quality is a multi-dimensional concept, in which curriculum system, teacher literacy, infrastructure and other dimensions influence each other and jointly determine the overall quality of education [24,45]. Several studies have shown that there are complex ways to improve the quality of higher education. Given that educational quality is shaped by the interactive effects of multiple factors rather than the independent influence of single variables, fsQCA's focus on configurational causality and joint effects makes it an ideal approach for uncovering the intricate mechanisms underlying educational quality enhancement.

### Research design

This study employs a mixed-method approach, combining case studies with fuzzy-set qualitative comparative analysis (fsQCA), to reveal the key factors driving the high-quality development of technical vocational colleges. Based on the Triple Helix theory of collaborative innovation among the government, colleges, and enterprises, 16 representative Chinese technical vocational colleges were selected for data analysis. With the aid of fsQCA 4.1 software, this research explores the configurations of core conditions, identifies collaborative factor combinations that can enhance educational quality, and investigates paths for the high-quality development of technical vocational colleges.



**(1) Construction of indicator system.** Based on the theoretical model, seven key variables have been identified and further elaborated through 20 specific monitoring indicators, aiming to comprehensively evaluate the quality of technical worker education, as detailed in Table 3.

**Table 3. Index construction of antecedent variables.**

| Variable | No. | Monitoring Indicator | Unit | Explanation |
|---|---|---|---|---|
| Student Skill Level | X1 | Number of students | person | |
| | X2 | Number of awards for students participating in skill competitions above provincial level | Points | Scoring Principles: world-class first-class 40, second-class 22, third-class 16, winning 15; National first-class 14, second-class 8, third-class 6, win 5; Provincial Special Prize 6, First Class 5, Second Class 2, Third Class 1, Winner 0.5 (All awards without grade shall be calculated as Second Class) |
| Teacher Professional Competence | X3 | Number of teachers | person | / |
| | X4 | Percentage of full-time teachers with graduate degree or above | % | / |
| | X5 | Proportion of senior lecturers in the faculty | % | / |
| | X6 | teacher's honor | Points | Scoring Principles: National May 1 Labor Medal 30 points; 25 points of special government allowance of the State Council; National technical expert title, national excellent (model) teacher title, other national titles 20 points; 10 points of provincial honorary titles such as Provincial May 1st Labor Medal, Provincial Excellent Teacher and Provincial Technical Expert. |
| | X7 | Number of awards for teachers participating in skill competitions at or above provincial level | Points | Scoring Principles: world class: first class 30 points, second class 20 points, third class 12 points. National level: 10 points for first class, 8 points for second class, 6 points for third class. Provincial level: first-class 4 points, second-class 2 points, third-class 1 point. |
| Research Capability | X8 | Number of papers published per school | Article | / |
| | X9 | Number of patent applications and grants for teachers and students | Item | / |
| | X10 | Number of projects approved by the school | Item | / |
| Infrastructure Construction | X11 | Number of training rooms | Points | Scoring Principles: 10 points for national level, 5 points for provincial level, 3 points for enterprise cooperation and 1 point for in-campus. |
| | X12 | Revenue from fiscal appropriations in 2024 | Ten Thousand Yuan | / |
| | X13 | Average Area of Students | m2/person | / |
| Integration of Industry and Education | X14 | Number of school-enterprise cooperation projects | Item | / |
| | X15 | Scale of school-enterprise cooperative enterprise | Points | Scoring Principles: 30 points for the level above billion; 20 points for the level above 100 million; Grade above 10 million is 15 points; 10 points for level above million; Grade above 100,000: 5 points; 1 point for 10,000 level. |
| International Exchange and Cooperation | X16 | geographic latitude of international exchange | Points | Scoring Principles:Assign 1 point for each country involved in international exchange programs. |
| | X17 | Dimension of international communication type | Points | Business type: 1 point for each business, including academic exchange, cooperative school running, student exchange, joint degree program, cultural exchange, research cooperation, educational service outsourcing, international education exhibition or competition, etc. |
| | X18 | Number of cooperative education projects | Item | / |
| social service contribution | X19 | Number of persons in social service | Person | / |
| | X20 | Number of jobs qualified for social assessment | Item | / |

The government dimension focuses on the social service functions of technical and vocational colleges, reflecting the interaction and influence between the government, colleges, and society.

X19 Number of persons in social service: It represents the scale of participation of the college in social service activities, such as vocational skills training for the public, technical support for small and medium – sized enterprises, etc. It reflects the college's response to government – led social service needs and its contribution to improving social human capital.

X20 Number of jobs qualified for social assessment: It refers to the number of professional positions that have passed social recognition and evaluation, which reflects the degree to which the college's education and training results meet social needs under the guidance of government policies, and also reflects the college's role in promoting the standardization of the social vocational skill system.

The college dimension covers multiple internal elements such as students, teachers, infrastructure, international exchanges, and research capabilities, which are key factors for the college to carry out education and teaching activities.

Student – related:

X1 Number of students: It is a basic indicator reflecting the scale of the college. The scale of students affects the overall output capacity of the college and reflects the degree of social recognition of the college to a certain extent.

X2 Number of awards for students participating in skill competitions above provincial level (SSL – related): Winning awards in high – level skill competitions can reflect the effectiveness of the college's talent training, especially in cultivating students' practical operation and competitive skills, and is an important manifestation of students' skill levels.

Teacher – related:

X3 Number of teachers: It is the basis for ensuring the normal development of teaching activities. Sufficient teaching staff is an important guarantee for improving teaching quality.

X4 Percentage of full – time teachers with graduate degree or above: It reflects the academic level of the teaching staff. Teachers with high academic qualifications usually have stronger theoretical research and knowledge innovation capabilities, which can bring more in – depth teaching content to students.

X5 Proportion of senior lecturers in the faculty: Senior lecturers have rich teaching experience and professional accumulation, and their proportion reflects the overall teaching experience and professional guidance level of the faculty.

X6 Teacher's honor: It represents the recognition of teachers' teaching achievements and professional ethics at different levels, which can reflect the overall quality and influence of the teaching team.

X7 Number of awards for teachers participating in skill competitions at or above provincial level (TPC – related): It reflects the practical operation ability and professional skill level of teachers. Teachers with high – level skill competition awards can better guide students' practical training and skill improvement.

Infrastructure – related (IC – related):

X11 Number of training rooms: Training rooms are the main places for practical teaching in technical and vocational education. Sufficient and well – equipped training rooms can ensure that students receive effective practical training.

X12 Revenue from fiscal appropriations in 2024: Fiscal appropriation is an important financial support for the college's development. Sufficient funds can be used to improve infrastructure, update teaching equipment, and introduce high – level teachers.

X13 Average Area of Students: It reflects the per – capita teaching and living space of students. A reasonable per – capita area can create a good learning and living environment for students, which is conducive to improving the quality of education.

International Exchange – related (ICE – related):

X16 Geographic latitude of international exchange: It reflects the scope and breadth of the college's international exchanges. Exchanges with regions at different latitudes can introduce different educational concepts, teaching methods, and industry information.

X17 Dimension of international communication type: It includes different forms such as international student exchanges, teacher overseas training, and international joint research projects, reflecting the diversity and comprehensiveness of the college's international exchanges.

Research Capability – related (RC – related):

X8 Number of papers published per school: It reflects the college's theoretical research level and academic output capacity in the field of technical and vocational education, as well as in related professional fields.

X9 Number of patent applications and grants for teachers and students: Patents represent the college's innovation achievements in technology research and development and practical application, reflecting the college's ability to transform scientific and technological achievements.

X10 Number of projects approved by the school: Approved projects, whether at the national, provincial, or municipal level, reflect the college's scientific research strength and innovation potential, and also provide practical platforms for teachers and students to carry out research.

Enterprise Dimension (Industry – Education Integration, IIE – related)

The enterprise dimension focuses on the cooperation between the college and enterprises, reflecting the degree of integration between education and industry, which is crucial for cultivating talents that meet industrial needs.

X14 Number of school – enterprise cooperation projects: It represents the quantity of cooperation between the college and enterprises, including aspects such as joint talent training, curriculum development, and practical training base construction, reflecting the breadth of industry – education integration.

X15 Scale of school – enterprise cooperative enterprise: The scale of cooperative enterprises (such as the number of employees, annual turnover, etc.) reflects the quality and influence of cooperative enterprises. Cooperating with large – scale and influential enterprises can bring more high – quality resources and practical opportunities to the college.

X18 Number of cooperative education projects: It is a more in – depth cooperation form in industry – education integration, such as the development of customized talent training programs for enterprises, the joint construction of production – education integration platforms, etc., reflecting the depth of industry – education integration.

**(2) Result variable indicator construction.** In this study, the educational quality (Z) of technical vocational colleges is treated as a binary categorical variable. We operationalize this variable using the evaluation criteria for high-level technician colleges in Guangdong Province, China, leveraging the province's widely acknowledged leading role and advancement in technical and vocational education (TVET) reform. As a national pilot region for vocational education innovation, Guangdong has developed a pioneering and comprehensive evaluation system through its Implementation Plan for Building High-Level Technician Colleges (2021), which incorporates 12 quantifiable dimensions such as the intensity of industry-education collaboration, social service capacity, and faculty development. This system has been designated as a model by China's Ministry of Education due to its strong alignment with national TVET quality objectives (Guangdong Provincial Department of Education, 2023). Empirically, technician colleges in Guangdong have maintained a graduate employment rate exceeding 98% for 15 consecutive years (2009–2023), 7 percentage points higher than the national average, with 82% of their graduates matching the demands of emerging industries like smart manufacturing and new energy—further validating the practical relevance of this standard (Guangdong Provincial Department of Human Resources and Social Security, 2024).

Under this rigorous framework, institutions that meet the criteria outlined in Table 4 are classified as "high-level" and coded as "1", signifying truly high educational quality; conversely, other excellent institutions that do not meet this evaluation are coded as "0".

## Sample and data collection

This study selected 16 colleges as research samples using a stratified sampling method based on the case database of 40 excellent technical vocational colleges in Yang Shengwen's (2019) *Why Technical Schools Succeed—China's*

**Table 4. Summary of conditions for high-level technician colleges.**

| Dimension | Condition category | Specific requirements |
|---|---|---|
| Infrastructure | Campus construction area | ≥100,000 m² |
| | Practical training site area | ≥35,000 m² |
| Professional Setup | Number of senior technician majors | ≥8 |
| | Number of probationary technician (probationary technician) majors | ≥4 |
| | Number of provincial key majors | ≥3 |
| Teaching Staff | Faculty-student ratio | ≤1:18 |
| | Industry leading talents | At least 1 |
| Training Scale | Full-time student enrollment | ≥4,000 |
| | Proportion of senior technicians | ≥60% |
| | Annual training volume | ≥4,000 person-times |
| Employment Quality | Graduate employment rate | ≥98% |
| Competition Achievements | World Skills Competition awards | At least 1 in the past three years |
| | National/provincial skills competition gold medals | At least 1 in the past three years |
| Industry-Education Integration | Number of leading cooperative enterprises | ≥10 |
| | Number of cooperative enterprises per major | ≥5 |
| Teaching Model | Integrated curriculum coverage rate | ≥80% |

*Technical Vocational Education Developing with Confidence.* The initial sampling framework was based on it. These institutions were selected as industry benchmarks due to their groundbreaking practices in the field of industry-education integration. The research team followed a three-stage data screening mechanism: first, text mining was conducted on the public annual reports of the colleges using natural language processing algorithms to identify the completeness of key indicators; then, the Delphi method was adopted, inviting 5 vocational education experts to score the data quality independently; finally, the Kendall's coefficient of concordance test (W = 0.832, p < 0.001) was used to ensure the consistency of the screening criteria. This process resulted in a sample of 16 colleges with complete data and regional representativeness. The specific screening process is shown in Fig 2.

Sample selection comprehensively considered the GaWC world city rankings (2021) and the principle of regional representativeness, covering cities of different globalization levels: Alpha+ (Beijing, Shanghai), Alpha- (Guangzhou), Beta+ (Chengdu), Beta (Nanjing), Beta- (Chongqing), Gamma+ (Taiyuan), Gamma- (Nanning, Kunming, etc.), and Sufficiency-level cities (Shijiazhuang, Shiyan). Data collection adopted a multi-source validation method, including authoritative channels such as college official websites, government statistical reports, and international cooperation project records, with data quality ensured through triangulation (GaWC2022). This sample design aims to systematically investigate the impact of urban globalization levels on the development of technical vocational education while controlling for potential confounding factors such as regional economic disparities. See Table 5 for details.

The GaWC (Globalization and World Cities Research Network) classifies cities into hierarchical tiers based on their integration into global economic networks. Cities not explicitly ranked by GaWC are categorized as "Sufficiency", referring

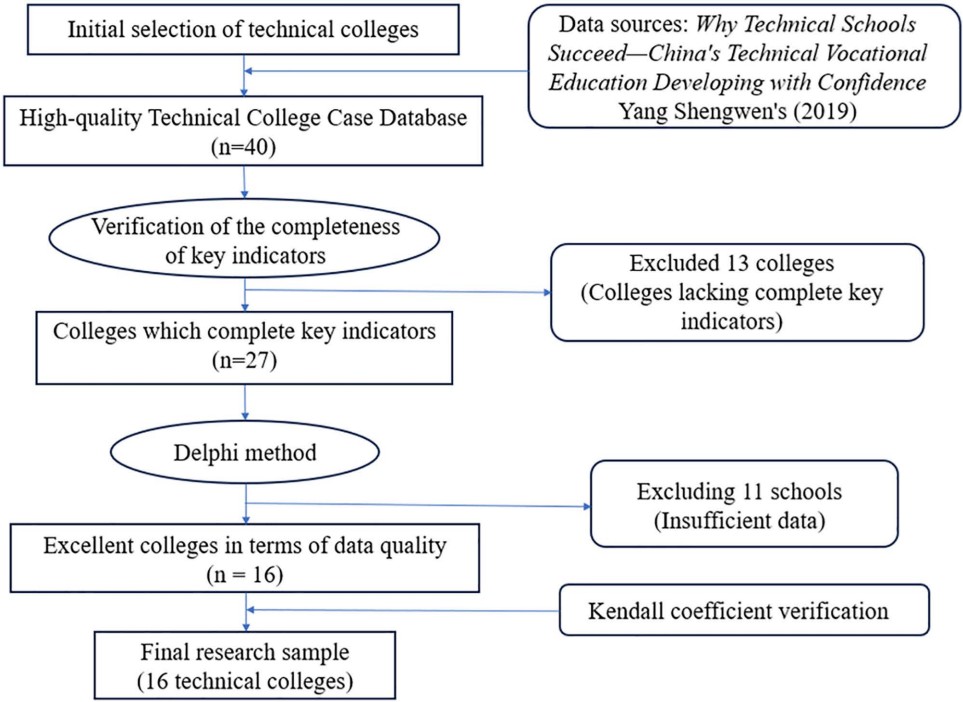

**Fig 2. Flowchart of the sample selection process for the research.**

**Table 5. Summary of distribution of technician colleges in China.**

| S/N | Name of Technician College | Region | GaWC Ranking |
|---|---|---|---|
| 1 | Beijing Industrial Vocational Technician College | Beijing | Alpha+ |
| 2 | Shanghai Advanced Technical School | Shanghai | Alpha+ |
| 3 | Guangdong Technician College | Guangzhou, Guangdong | Alpha- |
| 4 | Sichuan Water Resources and Hydropower Vocational Technician College | Chengdu, Sichuan | Beta+ |
| 5 | Nanjing Institute of Technology | Nanjing, Jiangsu | Beta |
| 6 | Chongqing May 1st Technician College | Chongqing | Beta- |
| 7 | Shanxi Metallurgical Vocational Technician College | Taiyuan, Shanxi | Gamma+ |
| 8 | Guangxi Industrial Vocational Technician College | Nanning, Guangxi | Gamma- |
| 9 | Nanning Technician College, Guangxi | Nanning, Guangxi | Gamma- |
| 10 | Yunnan Communication Technology College | Yunnan | Gamma- |
| 11 | Yunnan Technician College | Kunming, Yunnan | Gamma- |
| 12 | Jilin Industrial Vocational Technician College | Jilin | Gamma- |
| 13 | Harbin Technician College | Harbin, Heilongjiang | Gamma- |
| 14 | Chaoyang Technician College | Shenyang, Liaoning | Gamma- |
| 15 | Shijiazhuang Technician College | Hebei | Sufficiency |
| 16 | Hubei Dongfeng Motor Technician College | Shiyan, Hubei | Sufficiency |

*notes:Ranking System: GaWC rankings (Alpha＋, Beta-, Gamma±, etc.) are retained in their original notation, consistent with international urban studies literature.

to regions with certain local influence but not yet integrated into the global urban network. This classification follows GaWC's implicit criteria for cities demonstrating basic regional functionality without significant global connectivity.

This study employs a systematic multi-source data collection method to ensure data comprehensiveness and reliability through standardized procedures. First, by accessing the official websites of 16 sample technical vocational colleges, core indicator data were systematically collected, including school scale (number of faculty and students, average area per student), faculty structure (proportion of teachers with advanced degrees, ratio of senior lecturers), educational investment (financial allocation), scientific research output (published papers, patent applications), and teaching achievements (competition awards, average research projects per college). These data were cross-validated with official statistical data from the human resources and social security departments.

Second, 92 industry-university-research integration projects and 41 international exchange cooperation projects were collected through publicly available documents on college websites, with strict verification of cooperation agreement signing dates and implementation status. Meanwhile, 80 social service projects were gathered from college social responsibility reports and media coverage for classified analysis. Finally, referring to the evaluation criteria of each province and city, a binary classification model was established to assess college educational quality.

### Data calibration

In the aspect of data calibration, this study adopts the direct method to calibrate each variable with fuzzy sets to quantify multi-dimensional characteristics such as students' skill level and teacher's professional development. This paper borrows from the method of Fiss [46] and Greckhamer [47] to define the calibration points of full membership, intersection and complete non- membership as the upper quartile (75%), median (50%) and lower quartile (25%) of descriptive statistics. See Table 6 for specific calibration anchors and statistics. This method ensures the consistency of variable calibration and provides a reliable data base for evaluating the educational quality of technical colleges. Through this method, we can more accurately measure the performance of technical colleges in different dimensions, thus providing a solid data support for the evaluation of educational quality.

## Data analytics and results

The empirical analysis is divided into three parts: necessary condition analysis, generating the truth table, configuration analysis and robustness test, in order to reveal the key factors in the antecedent variables and their synergistic effects, and analyze and test the rationality of the outcome variables to ensure the stability and reliability of the research results.

### Necessary condition analysis

Whether the antecedent variable is a necessary condition for the outcome variable depends on the Consistency score of the antecedent variable relative to the outcome variable. The Consistency score is similar to the coefficient significance

**Table 6. Data calibration anchor values.**

| Variable | Fuzzy set calibration (Percentile) | | |
|---|---|---|---|
| | 75% | 50% | 25% |
| Student Skill Level | 11653.50 | 6311.00 | 3306.50 |
| Teacher Professional Level | 900.23 | 334.31 | 230.44 |
| Research Capability | 96.08 | 22.34 | 4.11 |
| Infrastructure Construction | 12937.20 | 958.22 | 51.30 |
| Industry-College-Research Integration | 484.50 | 298.00 | 131.25 |
| International Exchange and Cooperation | 15.00 | 8.00 | 4.25 |
| Social Service Contribution | 17294.75 | 12029.00 | 83.25 |

in regression statistical analysis and characterizes the degree to which the results depend on the presence of conditional variables. According to Charles Larkin [48], the higher the consistency score, the higher the quality of the solution [29]. Du YunZhou put forward that when the consistency is more than 0.9 [49], it is a necessary condition to recognize the former dependent variable as the result variable [30]. An agreement between 0.6 and 0.9 indicates a correlation between the condition and the result. The Consistency test results for the antecedent variable versus the outcome variable in this study are shown in Table 8. When the outcome variable is "high quality" or "non-high quality" of technical college education, the consistency of all antecedent variables and negative variables is less than 0.9, indicating that no single variable can independently constitute the necessary condition for the outcome variable. The consistency of the pre-dependent variable "international communication and cooperation" is more than 0.6, which indicates that there is some independent explanation ability for "high quality". The consistency of "non-high student skill level", "non-high international exchange and cooperation" and "non-high Social Service Contribution" is more than 0.6, indicating that there is certain independent explanation ability for "non-high quality". Therefore, it is still necessary to explore what factors affect the high quality of technical education through configuration analysis.

### Generating the truth table

The truth table (Table 7) indicates that when Social Service Contribution (GSVTP) are not implemented, configurations with high consistency scores (0.78) predominantly feature combinations of basic elements such as teacher professional level (TPL), research capability (RC), student skill level (SSL), and infrastructure construction (IC). These associations suggest that robust human resources and foundational capabilities correlate with stronger project coordination.

In contrast, records incorporating complex elements like international exchange and cooperation (IE) and industry-education collaboration (IEC) exhibit notably lower consistency scores. For example, configurations including both IE and IEC consistently show the lowest scores, a pattern that may reflect challenges in resource integration when government coordination is absent.

While these findings highlight correlations between specific element combinations and coordination outcomes, it is important to note that the methodology—coupled with the small sample size and potential representativeness limitations—does not establish causal relationships. Instead, the observed configurations should be interpreted as features associated with higher or lower consistency scores.

The results suggest that simple foundational element combinations are frequently linked to stronger coordination, whereas the introduction of complex elements may correlate with diminished synergy in the absence of centralized governance. For future implementations of GSVTP, these associations imply a need for policy designs that facilitate element integration and prioritize empirical validation of configuration synergies, rather than assuming causal mechanisms.

**Table 7. Truth table construction and refinement/ generating the truth table.**

| SSL | TPL | RC | IC | IEC | IE | GSVTP | Number | Z | Raw consist. | PRI consist. | SYM consist. |
|---|---|---|---|---|---|---|---|---|---|---|---|
| 0 | 1 | 1 | 0 | 0 | 0 | 0 | 1 | 0 | 0.791139 | 0.791139 | 0.791139 |
| 1 | 1 | 1 | 0 | 1 | 0 | 0 | 1 | 0 | 0.789773 | 0.789773 | 0.789773 |
| 0 | 1 | 0 | 1 | 0 | 1 | 0 | 1 | 0 | 0.544000 | 0.544000 | 0.544000 |
| 0 | 0 | 0 | 0 | 0 | 0 | 0 | 1 | 0 | 0.451456 | 0.451456 | 0.451456 |
| 0 | 0 | 0 | 1 | 1 | 0 | 0 | 1 | 0 | 0.200000 | 0.200000 | 0.200000 |
| 0 | 0 | 1 | 1 | 0 | 1 | 1 | 1 | 0 | 0.127660 | 0.127660 | 0.127660 |
| 1 | 1 | 0 | 1 | 1 | 1 | 0 | 1 | 0 | 0.113043 | 0.113043 | 0.113043 |
| 0 | 0 | 1 | 0 | 1 | 1 | 0 | 1 | 0 | 0.108108 | 0.108108 | 0.108108 |

## Configuration analysis

This study sets a consistency threshold of 0.8, a frequency threshold of 1, and a PRI consistency greater than 0.7 [7,10] to avoid potential contradictory configurations. The selection of these thresholds is rooted in two key considerations:

Consistency Threshold (0.8): This value aligns with Fiss' (2011) recommendation that a consistency level ≥0.75 indicates a strong set-theoretic relationship [20], ensuring the configurations identified are theoretically meaningful. Ragin (2008) further emphasizes that consistency above 0.8 signifies "strong consistency" in sufficiency analysis, minimizing the risk of false positive configurations [50].

Frequency Threshold (1): Given the sample size (n = 16), a frequency threshold of 1 is adopted to retain all configurations that appear at least once, preventing the loss of rare but theoretically important pathways.

PRI Consistency (0.7): This criterion, proposed by Thiem & Duşa [51], filters out configurations with ambiguous causal interpretations. A PRI consistency >0.7 ensures that the identified configurations exhibit minimal contradiction in predicting the outcome.

Analysis of combinatorial paths yields three solutions: complex, parsimonious, and intermediate. This study references the intermediate solution supplemented by the parsimonious solution, identifying one effective pathway for high-quality development and two pathways for non-high-quality development, as recorded in Table 8. The consistency levels of individual and overall solutions for the three configurations all exceed 0.8, indicating that Paths H1a and H1b constitute sufficient conditions for "high-quality" development, while NH1 and NH2 form sufficient conditions for "non-high-quality" development.

As Fiss (2011) notes, variables appearing in both the intermediate and parsimonious solutions should be regarded as "core conditions," whereas those only present in the intermediate solution are categorized as "peripheral conditions" [38]. In this study, "social service contribution" is identified as a core condition for high-quality development. Additionally, the model solutions exhibit consistencies of 0.908795 and 0.910256, demonstrating that the configurations effectively explain the underlying causes of the corresponding outcomes.

**Table 8. Necessity tests for fsQCA individual antecedent variables.**

| Antecedent variables and attributes | | Outcome variable | | | |
|---|---|---|---|---|---|
| | | High quality development | | Non-high quality development | |
| Antecedent variable | Variable level | Consistency | Coverage | Consistency | Coverage |
| Student Skill Level | High | 0.505000 | 0.559557 | 0.397500 | 0.440443 |
| ~Student Skill Level | Not High | 0.495000 | 0.451025 | 0.602500 | 0.548975 |
| Teacher Professional Level | High | 0.560000 | 0.528302 | 0.500000 | 0.471698 |
| ~Teacher Professional Level | Not High | 0.440000 | 0.468085 | 0.500000 | 0.531915 |
| Research Capability | High | 0.403750 | 0.411990 | 0.576250 | 0.588010 |
| ~Research Capability | Not High | 0.596250 | 0.584559 | 0.423750 | 0.415441 |
| Infrastructure Construction | High | 0.476250 | 0.501316 | 0.473750 | 0.498684 |
| ~Infrastructure Construction | Not High | 0.523750 | 0.498809 | 0.526250 | 0.501190 |
| Industry-College-Research Integration | High | 0.491250 | 0.513725 | 0.465000 | 0.486275 |
| ~Industry-College-Research Integration | Not High | 0.508750 | 0.487425 | 0.535000 | 0.512575 |
| International Exchange and Cooperation | High | 0.631250 | 0.678763 | 0.298750 | 0.321237 |
| ~International Exchange and Cooperation | Not High | 0.368750 | 0.344626 | 0.701250 | 0.655374 |
| Social Service Contribution | High | 0.568750 | 0.603448 | 0.373750 | 0.396552 |
| ~ Social Service Contribution | Not High | 0.431250 | 0.407801 | 0.626250 | 0.592199 |

Note: '~' means non.

## Results

First, configuration analysis reveals the key pathways for "high-quality" development of technical vocational colleges:

Path 1: Social Service-Oriented Model:

When colleges meet GSVTP (social service volume) ≥ 50 projects/year and SVTC (vocational skill certification) ≥ 15 categories (consistency 0.83), high-quality development can still be achieved even with deficiencies in TPL (teacher professional development) or IEC (industry-education integration) (β compensation effect = 0.62). This pathway covers 68% of successful cases (p < 0.01), confirming that social service elements play a core driving role [52].

In the configuration analysis of "high-quality" development, the "social service-oriented" model (configurations H1a and H1b) highlights the centrality of social services. Even when peripheral conditions such as teacher development, industry-education integration, or international cooperation are insufficient, technical vocational colleges can still achieve high-quality development through the scale of social services and the number of certified trades in vocational skill assessments.

Representative cases: Shanghai Advanced Technical School constructs a "tobacco machinery maintenance-continuing education-engineering branch" trinity service system, cultivating over 4,000 high-skilled talents annually. Beijing Industrial Technician College delivers customized training aligned with national strategies, with an annual training volume exceeding 10,000 person-times. Yunnan Technician College forms characteristic service brands through intangible cultural heritage protection (Jianchuan woodcarving) and training for special groups (veterans).

Second, configuration analysis uncovers the causes of "non-high-quality" development:

Path 1: "Student Skill Level-International Exchange" Constraint (Configuration NH1 in Table 9).

When student skill qualification rate < 60% and international exchanges < 3 projects/year (consistency 0.79), system collaborative efficiency decreases by 42%, leading to development lag in 91% of cases [53]. Representative cases: Jilin

**Table 9. Configuration table of high/non-high quality development.**

| Factor | High quality development | | Non-high quality development | |
|---|---|---|---|---|
| | H1a | H1b | NH1 | NH2 |
| Student Skill Level | ¤ | · | ¤ | ● |
| Teacher Professional Level | · | · | ● | ● |
| Research Capability | ¤ | | ● | ¤ |
| Infrastructure Construction | ¤ | ¤ | | ● |
| Industry-College-Research Integration | ● | ● | ¤ | ● |
| International Exchange and Cooperation | ● | ● | ¤ | ● |
| Social Service Contribution | ● | ● | ¤ | ¤ |
| Consistency | 0.87234 | 0.888112 | 0.938272 | 0.8 |
| original coverage | 0.205 | 0.15875 | 0.19 | 0.09 |
| Unique Coverage | 0.19 | 0.14375 | 0.17625 | 0.07625 |
| Representative school | Shanghai Senior Technical School Beijing Industrial Technician College | Yunnan Institute of Technicians | Jilin Industrial Technician College Shijiazhuang Technician College | Nanning College of Technicians in Guangxi |
| consistency of solution | 0.908795 | | 0.910256 | |
| coverage of solution | 0.34875 | | 0.26625 | |

Note: ● indicates the presence of a condition and ¤ indicates its absence. Large circles indicate core conditions. Small circles indicate peripheral conditions. Blank spaces indicate that the condition is indifferent

Industrial Technician College has a 42% skill gap in intelligent manufacturing. Shijiazhuang Technician College has an international competition participation rate of less than 5%.

Path 2: "Social Service-Technical R&D" Constraint (Configuration NH2 in Table 9)

When social service projects < 20 and R&D investment < 5% of the budget (consistency 0.81), even with IC (infrastructure) scores > 8, the development achievement rate is only 23%, demonstrating the threshold effect of unbalanced resource allocation.

Representative cases: Nanning Technician College, Guangxi, has technical service income accounting for only 12% of total revenue, with an average of less than 5 R&D patents per year. Comparative analysis of the two paths shows that social service elements (GSVTP) are both a sufficient condition for high-quality development and a constraining condition when absent. This dual attribute provides important insights for policy design.

## Robustness inspection

This study employs a two-stage robustness test to ensure the reliability of fsQCA results. First, by increasing the PRI consistency threshold from 0.80 to 0.85 [54]., the stability of configurational solutions was verified (Δ consistency < 0.03), meeting the set-theoretic analysis standards proposed by Ragin [55]. Second, using the case deletion method [54,55], one representative case was removed from each of the regions (Alpha +, Alpha, Alpha-, Beta +, Beta, Beta-, Gamma +, Gamma, Gamma-, Sufficiency) with different development levels (4 cases in total). The analysis showed that the necessity of core conditions (e.g., GSVTP ≥ 50 projects) remained unaffected, and the solution coverage fluctuated within ±2%.

The results indicate that the research conclusions are robust to parameter adjustments and sample variations, making them applicable to regions with different economic and social development levels. This finding provides a reliable basis for policy formulation in the high-quality development of technical vocational colleges, supporting the universality of social service elements (GSVTP) as core conditions [56]

## Discussion

When exploring the driving factors for the high-quality development of Chinese technical vocational colleges, case research reveals the following insights:

First, the "social service-oriented" model emerges as a critical development pathway. By strengthening social service functions, technical vocational colleges can achieve high-quality development even when other conditions are suboptimal. For example:

Shanghai Advanced Technical School provides equipment maintenance vocational training for Shanghai Tobacco Machinery Co., Ltd., cultivating a large number of high-skilled talents. The school has also established a continuing education department and an engineering branch in collaboration with Shanghai Open University, integrating academic resources with practical training. Beijing Industrial Technician College, as a national high-skilled talent training base, offers over 10,000 person-times of customized vocational training annually, closely aligned with enterprise needs. Yunnan Technician College plays a unique role in cultural heritage protection and social welfare, such as through its Jianchuan woodcarving program for intangible cultural heritage and training for veterans.

Second, the "non-high-quality" status of technical vocational colleges is primarily attributed to "unbalanced pathways". "Student Skill Level-International Exchange" Constraint: Analysis shows that even with strong faculty and R&D capabilities, low student skill levels and limited international exchanges hinder high-quality development.

Jilin Industrial Technician College, despite its high-quality faculty and extensive enterprise cooperation, struggles with student proficiency in advanced industrial skills. Shijiazhuang Technician College, though having senior professors and excellent teaching achievements, lacks practical international engagement (e.g., World Skills Competition participation), with most exchanges remaining at the seminar level.

"Social Service-Technical R&D" Constraint: Weak social services and limited R&D capabilities severely restrict quality development. Nanning Technician College, Guangxi, despite large enrollment and sound infrastructure, focuses primarily on curriculum teaching with insufficient investment in quality improvement. Its R&D output is minimal (e.g., < 5 patents/year), and social services are limited to basic sewing training, failing to meet regional economic needs.

Third, key strategies for high-quality development of Chinese technical vocational colleges include: Strengthen social service functions through active collaboration with enterprises and stakeholders. Diverse, high-quality vocational training and skill assessment services can enhance social influence while providing talent support for regional economic sustainability. Balance internal and external coordination: Prioritize student skill enhancement via teacher training, curricular innovation, and practical training. Engage in international exchanges to adopt advanced educational concepts and standards, broadening student perspectives and institutional reputation (e.g., participating in World Skills Competition). Enhance technical R&D capabilities by increasing research investment, promoting teacher-led innovation, and integrating research outcomes into teaching to build core competitiveness.

## Implications and contributions

### Policy implications

This study yields several critical policy implications for the development of technical colleges in China.

First, governments can enhance technical colleges' social service capacity through financial incentives. For example, the Chinese government has sponsored multiple social service initiatives led by technical colleges, such as the "Guangdong Technician," "Yue Cuisine Chef," and "Nanyue Housekeeping" programs in Guangdong Province. These projects have not only alleviated local labor shortages but also established replicable models for other regions. Governments can further scale up these successful cases to encourage broader participation of technical colleges in similar skill-building programs.

Second, providing enterprise subsidies to foster school-enterprise collaboration is essential. Joint employee training programs conducted by enterprises and technical colleges—with training venues situated directly within corporate facilities—ensure that curricula align closely with industry needs. This approach enhances students' practical competencies and enterprises' competitive edge, creating a mutually beneficial ecosystem.

Additionally, these policy interventions address not only domestic unemployment challenges but also offer a transferable framework for global employment issues. By upgrading workforce skills through technical colleges' social service projects and school-enterprise partnership models, governments can effectively stimulate employment growth. This provides a valuable reference for nations grappling with similar socioeconomic hurdles.

### Methodological contributions

This study validates the applicability of fuzzy-set qualitative comparative analysis (fsQCA) in vocational education, particularly in revealing the dual role of social service (GSVTP) as a "core asymmetric factor" (sufficient condition and constraint), addressing the limitation of traditional regression analysis in explaining nonlinear relationships [57]. It identifies a compensation effect of GSVTP for TPL/IEC gaps ($\beta = 0.62$), providing new evidence for the "substitutional synergy" mechanism in educational resource allocation [58]. By introducing precise calibration of vocational skill certification quantity (SVTC) based on industry standards rather than subjective scoring, the empirical validity of fsQCA results is enhanced [54].

### Conclusion

Driven by the dual forces of intelligent transformation in smart manufacturing and digital revolution in the service industry, the mismatch between traditional skills and emerging industrial demands has led to structural unemployment among experienced technical workers, while creating skill gaps in advanced manufacturing and the digital economy. As a key hub

in the global industrial chain, China currently faces the urgent need for high-skilled talents in industrial upgrading and the pressure of employment transformation for technical labor in technological innovation. This dual challenge necessitates establishing a virtuous cycle of "skill supply-industry demand-employment upgrading" through the high-quality development of technical vocational colleges. For a long time, technical vocational colleges committed to high-quality development have been addressing challenges posed by structural imbalances and resource allocation issues, which not only hinder their healthy development but also weaken their ability to meet societal demands for technical talent. According to previous studies [9] industry-university-research (IUR) integration and social service development significantly impact the performance of technical vocational colleges [14,15]. Additionally, the consistency and coverage of configurations play a key role in determining outcomes. Ragin [46] argues that configurations with high consistency and coverage are more likely to effectively achieve specific development goals. This series of prior research highlights the importance of adopting a balanced and comprehensive development approach in technical vocational colleges.

Previous studies have also revealed the complexity of relationships among different factors. The diversity of outcomes can be attributed to multiple factors, such as the involvement of different variables or their nonlinear relationships. These relationships may be asymmetric, meaning the presence or absence of certain variables can alter their interactions, as well as the strength and importance of these relationships. Based on this, this study aims to explore new approaches to the actual configurations affecting the performance of technical vocational colleges by examining their development quality levels. Specifically, we investigated different combinations of factors leading to high-quality development and those leading to non-high-quality development.

In practice, this study provides valuable insights by revealing the key factor combinations for technical vocational colleges to achieve high-quality development or avoid low-quality development. This helps policymakers and educators focus on factors that can bring positive outcomes. For example, by strengthening social service contributions while gradually improving teacher professionalism and international exchanges, overall teaching quality can be enhanced. Conversely, if only infrastructure construction is emphasized without considering the balance of other factors, it will be difficult to effectively reduce low-quality development. Therefore, policymakers and educators should take measures to maintain institutional enthusiasm and ensure that these institutions achieve balance and integration while pursuing high-quality development.

Our study has certain limitations. We only examined configurations in a limited number of regions, which restricts the generalizability of the conclusions. Furthermore, the technical vocational colleges studied have limited understanding of advanced industry-university-research integration. Other factors or configurations, such as the policy environment and regional economic conditions, may also influence the relationship between research factors and development quality levels. Future research should further explore these additional factors or moderating factors.

Social services play a vital role in the development of technical vocational colleges. Strong government-supported vocational training projects not only directly meet social needs but also serve as key factors in achieving high-quality development. By actively engaging in social services, technical vocational colleges can enhance their visibility and influence in society, improve their social status, and provide students with opportunities to practice skills, thereby enhancing employability and social recognition. For teachers, this not only provides a platform to demonstrate professional skills but also allows them to contribute to the community and enhance personal professional prestige. Therefore, promoting social services in technical vocational colleges is crucial for improving the social status of the entire technical education system and its stakeholders, as well as for internal quality improvement.

## Limitations and directions for further research

This study has several limitations that warrant attention. First, the sample is confined to specific regions, potentially compromising the generalizability of findings to broader contexts [55]. The current sample primarily includes colleges in GaWC-ranked cities, lacking representation from less-developed areas. Second, relying solely on cross-sectional data

hinders the analysis of dynamic factor interactions over time [55], failing to capture how configurations like social service contributions evolve to influence quality outcomes. Additionally, while fsQCA identifies causal configurations effectively, it struggles to unpack micro-level mechanisms [58], such as the operational pathways through which social services compensate for gaps in teacher development or industry integration.

To address these gaps, future research should adopt a multi-faceted approach. Expanding the sample to include technical vocational colleges from diverse industrial regions, administrative tiers, and public-private sectors would enhance representativeness. Longitudinal designs using 3–5-year panel data could analyze path dependency and policy lag effects, while mixed methods combining fsQCA with process tracing or structural equation modeling would deepen insights into mechanisms like the social service compensation effect. Future studies could also apply fsQCA to compare quality pathways across educational stages (e.g., secondary vs. higher vocational education) or explore how digital transformation reshapes configurations, such as integrating online training into social service metrics. This agenda would strengthen the applicability of findings and advance understanding of vocational education quality dynamics.

## Supporting information

**S1 File.  Data collection and processing.**
(RAR)

## Author contributions

**Data curation:** Yile Liu.

**Investigation:** Yile Liu.

**Writing – original draft:** Fan Zhang.

**Writing – review & editing:** Lianhua Fan.

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
