## [Decision Letter · Decision Letter 0]

5 May 2025

PONE-D-25-07768How to Promote the Quality of Chinese Technical Education�A Fuzzy-set Qualitative Analysis of 16 Technical Colleges in ChinaPLOS ONE

Dear Dr. Zhang,

Thank you for submitting your manuscript to PLOS ONE. After careful consideration, we feel that it has merit but does not fully meet PLOS ONE’s publication criteria as it currently stands. Therefore, we invite you to submit a revised version of the manuscript that addresses the points raised during the review process.

We look forward to receiving your revised manuscript.

Kind regards,

Pengpeng Ye

Academic Editor

PLOS ONE

[This work was supported by a project grant from 2024 Guangdong Provincial Research Topic on Technical Education and Vocational Training: A Study on the Compatibility of Talent Cultivation Models in Guangdong Technical Colleges with Industrial Structure Upgrading and Transformation (KT2024052) ; 2024 National University and Vocational College Logistics Teaching Reform and Research Project: Research on Teaching Reform and Training Mode of Cross-border E-commerce and Cross-border Logistics Compound Talents in Technical Colleges Guided by OBE Concept (JZW2024262).]

 [The author(s) received no specific funding for this work]

[None].

Reviewers' comments:

Reviewer's Responses to Questions

**Comments to the Author**

1. Is the manuscript technically sound, and do the data support the conclusions?

Reviewer #1: Yes

Reviewer #2: Partly

Reviewer #3: Yes

Reviewer #4: No

Reviewer #5: Partly

Reviewer #6: No

Reviewer #7: No

Reviewer #8: Yes

2. Has the statistical analysis been performed appropriately and rigorously? 

Reviewer #1: Yes

Reviewer #2: N/A

Reviewer #3: No

Reviewer #4: I Don't Know

Reviewer #5: I Don't Know

Reviewer #6: No

Reviewer #7: No

Reviewer #8: Yes

3. Have the authors made all data underlying the findings in their manuscript fully available?

Reviewer #1: Yes

Reviewer #2: No

Reviewer #3: Yes

Reviewer #4: No

Reviewer #5: No

Reviewer #6: No

Reviewer #7: Yes

Reviewer #8: Yes

4. Is the manuscript presented in an intelligible fashion and written in standard English?

Reviewer #1: Yes

Reviewer #2: No

Reviewer #3: Yes

Reviewer #4: Yes

Reviewer #5: Yes

Reviewer #6: Yes

Reviewer #7: No

Reviewer #8: Yes

5. Review Comments to the Author

Reviewer #1: General recommendations:

I suggest authors expand the justification by clearly highlighting the relevance and original contribution of the research in relation to previous studies.

In the literature review section, authors should avoid unnecessary repetition and better integrate national and international studies into clear topics. Don't forget to correct incomplete or incorrect references.

It may be necessary to reduce narrative descriptions of cases, focusing only on key points and how they exemplify the configurations found.

Authors should expand on the study's practical contributions.

Reviewer #2: I do not recommend the publication of this manuscript due to fundamental issues in both content and structure. The major concerns are as follows:

1. From the introduction onwards, the manuscript suffers from a disorganized structure and weak logical flow. The argument is difficult to follow, and the narrative lacks a coherent progression typical of academic writing.

2. The topic, while generally relevant, is poorly framed and lacks originality. The research findings do not provide novel insights or actionable implications for advancing the field of technical education in China or internationally.

3. The literature review appears to be a mere aggregation of prior studies without synthesis or critical evaluation. It fails to establish a solid theoretical foundation for the study.

4. The use of fuzzy-set qualitative comparative analysis (fsQCA) is inadequately explained and poorly executed. The sampling rationale, variable selection, and calibration process lack transparency and rigor, which raises serious concerns about the validity of the results.

5. The results are presented as mechanical outputs of the fsQCA software, with minimal interpretation or theoretical reflection. The discussion does not go beyond descriptive repetition of configurations and fails to provide meaningful insights.

6. The reference formatting is inconsistent throughout the manuscript, with noticeable citation errors. This indicates a lack of attention to academic standards and detracts from the manuscript’s credibility.

In summary, the manuscript does not meet the basic criteria for publication in terms of structure, content, methodological rigor, and academic contribution. I recommend rejection.

Reviewer #3: Overall, I find the topic of the article interesting and relevant; however, substantial revisions are needed before it can be considered for publication. Below some suggestions:

a) While the introduction provides a comprehensive overview of the topic and relevant policy context, the specific academic contribution of the study remains unclear and should be more explicitly articulated. In my opinion the contribution is the weakest point of the paper.

b) The literature review, while comprehensive in scope, is largely descriptive and lacks the critical engagement and analytical depth necessary to meaningfully support the research. It reads more like a background summary than a foundation for an original academic contribution. To strengthen this section, the authors should critically synthesize the reviewed studies, identify gaps in the existing literature, and clearly formulate research hypotheses that articulate the study's intended scientific contribution.

c) The research design is generally conceptually relevant, but it requires a clearer and more systematic structure. The chosen methodology lacks sufficient justification, and alternative methodological approaches are not discussed, nor are their respective advantages and limitations. This omission weakens the analytical rigor and should be addressed as part of a more comprehensive robustness analysis, which is currently underdeveloped.

d) While the paper proposes several relevant directions for policy, the recommendations remain overly general and would benefit from clearer prioritization, contextual differentiation, and discussion of feasibility and implementation mechanisms.

Reviewer #4: Thank you for the opportunity. I have my comments in the attached document. My main criticism would be on the importance of the research question - why is it important to study this and how it addresses the gap in literature and the empirical rigor.

Reviewer #5: This study examines the factors that support “high quality” development of technician colleges in China. The authors use a fuzzy-set qualitative comparison analysis to examine characteristics of 16 selected technician colleges. This method numerically encodes information about the colleges across 7 domains and converts them scores representing degrees of membership within sets for each domain. Set theoretic analyses are then applied to identify conditions that are necessary or sufficient in configurations associated with high quality. The authors find strong social-service engagement is a core condition in the configuration of all high-quality technician colleges, and non-high-quality outcomes arise when colleges exhibit either (i) low student skill level and low international exchange, or (ii) low social-service activity and low technological R&D. Specific comments are listed below.

- The sample for analysis includes 16 technician colleges selected from a list of 40 colleges included in a book profiling “outstanding technician colleges.” Why were colleges selected from this book? If the intent is to identify configurations related to high quality, it seems strange to start from a sample of “outstanding” colleges, which limits the ability to distinguish high quality from low quality since they are all seemingly exceptional. Why don’t the authors use a sample that is representative of all colleges?

- The process for selecting the 16 colleges for analysis out of the population of 40 colleges is not clear. At first, it sounds like they are randomly selected after stratifying them into four regions. However, at the end of page 3, the authors say “After further data collection, 16 technician colleges were selected as the final research objects,” How were these 16 selected, what was the further data collection and how was it used to select them?

- Page 2, near the end of the second paragraph, there is a missing reference.

- The last paragraph in the section “(1) Sample and Data Collection” appears to duplicate the information in the preceding paragraphs.

- I think the authors should refrain from using language that implies causality. It does not seem like this method—combined with the small sample with issues of representativeness described above—can make causal inferences. Instead, configurations/characteristics should be described as associated with high quality.

- The description of the construction of antecedent variable indicators is confusing. It is not always clear how they are constructed, particularly for the indicators that are described as being in “minutes” in table 2. What does “minutes” mean here, and how are the points systems described in various rows of table 2 developed?

- I don’t understand the difference between “observation index” and “indicator” in table 2. It seems like observation index includes indicators, except when they need to be defined with some additional points system, which is then described in the indicator column.

- Many of the indicators are basically just proxies for the size of the college (e.g., number of students, number of teachers) or are highly influenced by size (e.g., number of papers published). Wouldn’t it be better to have indicators that are normalized by size, like a student-teacher ratio?

- Instead of using “human” as a unit in table 2, “persons” or “individuals” might be a better term.

- It is not clear how the binary outcome is constructed. Page 8 describes many indicators of high quality, but I do not understand how they are then translated into a binary indicator.

Reviewer #6: The introduction section of the article does not raise a scientifically meaningful question. The research lacks a central theme and focused research content.

The data in this study comes from secondary sources, specifically from a book. The sampling process and the characteristics of the sampled data are not described in detail, making it impossible to verify the authenticity and reliability of the data. Additionally, the overall workload of the study is clearly insufficient.

The selection of the indicator system lacks scientific rigor and is based on the author’s subjective judgment. Since the paper does not have a core research question, the author’s choice of indicators is arbitrary, rendering the evaluation scientifically unfounded. Moreover, the indicators exhibit endogeneity (interdependence), which is not thoroughly explained.

The empirical analysis is overly generalized and does not address the actual problem. It merely follows a routine evaluation method without exploring the formation mechanism of the scientific issue, causal analysis, or scientific pathways. This paper is merely a subjective indicator evaluation study without substantive theoretical or empirical contributions.

Reviewer #7: This paper examines technical colleges in China. There are several serious issues in the paper which I present below (in order of appearance in the paper) and which, after careful consideration of the paper, caused me to recommend a rejection.

Comments

1. General comment: I have significant doubts about the value added of the paper. After reading it I don’t understand what that value added is as the authors don’t indicate it. Additionally as the paper focuses on China the authors should draw some conclusions for the international audience but that does not happen in the paper.

2. General comment: the paper basically should be in large parts rewritten to make a good and interesting reading. As of now the narrative is quit poor, the paper in large parts uninteresting, very difficult to understand and makes a poor reading.

3. The abstract is very difficult to understand and should be completely rewritten.

4. The Introduction is very superficial. It lacks an indication what data and method will be used in the paper, what value added the paper has and what its structure is.

5. The Literature Review is superficial too. It is difficult to understand and simply boring.

6. Based on the description in the paper I have no idea why the authors have two parts dedicated to the description of the data sample: one on page 3 (Sample and Data Collection) and one on page 9 (Data Acquisition and Calibration)

7. Why is the first paragraph on page 4 a copy-paste of the last 4 paragraphs on the previous page?

8. The authors state on page 4 that they chose 40 cases of outstanding technician colleges. What is the reason behind this? All these all technician colleges in China? Or have the authors applied a procedure to select those 40 colleges? This totally unclear.

9. A similar comment concerns the 16 technician colleges selected as the final research objects.

10. The section Construction of antecedent variable indicators has to be completely rewritten. Most of its content can probably be presented in a Table.

11. The sections following the Construction of antecedent variable indicators section are difficult to understand and difficult to follow. I have no idea what the authors are trying to describe in those sectio

Reviewer #8: COMMENTS FOR MANUSCRIPT PONE-D-25-07768

1. The study presents the results of original research.

Yes

2. Results reported have not been published elsewhere.

To the best of my knowledge, the results in the report have not been published elsewhere.

3. Experiments, statistics, and other analyses are performed to a high technical standard and are described in sufficient detail.

The sampling method is not comprehensively described. Why are they choosing only high-performing colleges? There is also no justification on the number of colleges included in the study. It is also not clear why they selected the number of selected colleges differs across regions (for example, why did they choose 5 in the east, 2 in the middle, 6 in the west and so on?).

4. Conclusions are presented in an appropriate fashion and are supported by the data.

Paragraph 2 of the conclusion is supposed to be part of the discussion section, which is completely absent in the paper.

5. The article is presented in an intelligible fashion and is written in Standard English.

The language used in the article is flawless but will need some proof reading to remove the minor grammatical errors currently in the manuscript (for example, the first sentence of the theoretical model has some missing words). However, there is a lot of repetitions in the sampling section.

6. The research meets all applicable standards for the ethics of experimentation and research integrity.

Yes.

7. The article adheres to appropriate reporting guidelines and community standards for data availability.

Yes.

6. PLOS authors have the option to publish the peer review history of their article (what does this mean? ). If published, this will include your full peer review and any attached files.

**Do you want your identity to be public for this peer review?** For information about this choice, including consent withdrawal, please see our Privacy Policy .

Reviewer #1: **Yes: ** Raul Afonso Pommer Barbosa

Reviewer #2: No

Reviewer #3: No

Reviewer #4: No

Reviewer #5: No

Reviewer #6: No

Reviewer #7: No

Reviewer #8: No

---

## [Author Response · Author response to Decision Letter 1]

24 Jun 2025

We appreciate the time and effort that you and the reviewers dedicated to providing feedback on our manuseript and are grateful for the insightfulcomments on and valuable improvements to our paper.

---

## [Decision Letter · Decision Letter 1]

24 Jul 2025

PONE-D-25-07768R1How to Promote the Quality of Technical Education�A Fuzzy-set Qualitative Analysis of 16 Technical Colleges in ChinaPLOS ONE

Dear Dr. Zhang,

Thank you for submitting your manuscript to PLOS ONE. After careful consideration, we feel that it has merit but does not fully meet PLOS ONE’s publication criteria as it currently stands. Therefore, we invite you to submit a revised version of the manuscript that addresses the points raised during the review process.

**ACADEMIC EDITOR: ** After carefully considering the opinions from both rounds of review, I can see that you have addressed most of the concerns raised by the reviewers, and the overall quality of the work has improved significantly. To further enhance the quality of the manuscript and align it with the journal's requirements, I recommend adequately addressing the opinions of the two reviewers who suggested rejection. If you are unable to do so, please provide sufficient reasons and clearly mark these as limitations or future research directions in the manuscript. I believe that the manuscript will be a valuable contribution to our readership. 

We look forward to receiving your revised manuscript.

Kind regards,

Pengpeng Ye

Academic Editor

PLOS ONE

Journal Requirements:

Reviewers' comments:

Reviewer's Responses to Questions

**Comments to the Author**

1. If the authors have adequately addressed your comments raised in a previous round of review and you feel that this manuscript is now acceptable for publication, you may indicate that here to bypass the “Comments to the Author” section, enter your conflict of interest statement in the “Confidential to Editor” section, and submit your "Accept" recommendation.

Reviewer #1: All comments have been addressed

Reviewer #2: All comments have been addressed

Reviewer #3: All comments have been addressed

Reviewer #4: All comments have been addressed

Reviewer #5: (No Response)

Reviewer #7: (No Response)

Reviewer #8: All comments have been addressed

2. Is the manuscript technically sound, and do the data support the conclusions?

Reviewer #1: Yes

Reviewer #2: Yes

Reviewer #3: Yes

Reviewer #4: (No Response)

Reviewer #5: No

Reviewer #7: No

Reviewer #8: Yes

3. Has the statistical analysis been performed appropriately and rigorously? 

Reviewer #1: Yes

Reviewer #2: Yes

Reviewer #3: Yes

Reviewer #4: (No Response)

Reviewer #5: No

Reviewer #7: No

Reviewer #8: Yes

4. Have the authors made all data underlying the findings in their manuscript fully available?

Reviewer #1: Yes

Reviewer #2: Yes

Reviewer #3: Yes

Reviewer #4: (No Response)

Reviewer #5: No

Reviewer #7: Yes

Reviewer #8: Yes

5. Is the manuscript presented in an intelligible fashion and written in standard English?

Reviewer #1: Yes

Reviewer #2: Yes

Reviewer #3: Yes

Reviewer #4: (No Response)

Reviewer #5: No

Reviewer #7: No

Reviewer #8: Yes

6. Review Comments to the Author

Reviewer #1: (No Response)

Reviewer #2: The manuscript entitled “How to Promote the Quality of Technical Education? A Fuzzy-set Qualitative Analysis of 16 Technical Colleges in China” offers a timely, policy-relevant contribution to the TVET literature by scrutinising high-quality development pathways across a systematically selected national sample and thus fills an evident gap in empirical evidence on Chinese technician colleges.

Its central argument—that social-service capacity acts as the foundational condition for superior institutional performance—is developed through a clear, well-theorised fsQCA design that integrates Triple-Helix logic and delivers nuanced configuration results; this analytic strategy is both innovative and convincingly executed, enabling the study to move beyond single-factor explanations.

The prose is fluent and idiomatic, with terminology and syntax that read naturally to an international audience; transitions among Introduction, Literature Review, Methods and Findings are smooth, and the technical discussion of calibration anchors, truth-table thresholds and robustness checks demonstrates a solid command of set-theoretic methodology.

Conceptual originality is evident in the seven-dimension evaluation model that blends government, college and enterprise indicators and in the identification of two distinct developmental bottlenecks (“student skill–international exchange constraints” and “social service–technology R&D constraints”), both of which add explanatory depth to existing quality-assessment frameworks.

The discussion section thoughtfully links statistical configurations to concrete policy initiatives—such as SkillsFuture and provincial high-level-college criteria—thereby translating empirical insights into actionable recommendations for practitioners and decision-makers, while also acknowledging methodological limitations and future research directions.

Formatting, referencing and data-availability statements fully align with PLOS ONE guidelines; tables are clearly labelled, appendices are complete, and ethical-compliance declarations are explicit. In view of the study’s meaningful topic, lucid writing, well-substantiated innovation, thorough discussion and exemplary presentation, I recommend the manuscript for publication without further revision.

Reviewer #3: The authors have thoroughly addressed the vast majority of the comments raised in the previous round of reviews. The manuscript has significantly improved in terms of clarity, methodological rigor, and interpretability of results. believe the paper now meets the standards of the journal and should be accepted for publication.

Reviewer #4: (No Response)

Reviewer #5: The revisions have improved language clarity in parts and trimmed redundancy in the literature review. However, several methodological issues remain.

- I remain skeptical that we can learn anything from this study based on the sample. Only 16 colleges are included from the starting list of 40, and the text does not present much justification for why these 16 were chosen—based on what I can understand from the text and response reviewer comments, it seems these 16 were essentially a convenience sample based on whether the authors could gather enough data for the school. Since the 16 are already being chosen from a set of 40 “excellent” colleges, I do not think this small convenience sample can possibly offer reasonable representativeness of China’s TVET institutions, and the ability to distinguish high quality from within this sample seems extremely limited. The authors should clarify the exact inclusion criteria, justify why these 16 cases capture the heterogeneity of China’s TVET institutions, and discuss limits on generalizability. It would be helpful to add (i) a flow-chart of case inclusion, (ii) a descriptive table comparing the 16 to the 40 and to national averages. The authors should describe the analysis as exploratory an limit the broad, generalized claims accordingly.

- “High quality” is defined based on criteria from Guangdong province. Why were the criteria from this province chosen to apply to schools from across China? The authors should provide justification for this choice or examine/discuss the potential limitations from applying province-specific criteria for all schools.

- Many of the sub-indicators are translated into “points” with no description of where these point systems come from or validation. For data quality, the authors at one point report a coder reliability kappa coefficient of 0.87, but there is no description of the coding procedures, how many coders, etc.

- The calibration thresholds use percentiles, and there are only 16 schools. This seems like it would produce fairly arbitrary cutoffs that are very sensitive to adding/subtracting a school. Robustness checks would be helpful to assess how sensitive results are to the calibration thresholds.

- The authors acknowledge small-sample limitations but still draw strong causal conclusions (e.g., “social service constitutes the foundation of high-quality vocational education” in the Abstract). Again, I do not think we can learn anything causal from this analysis.

- Most inputs are scraped from institutional web sites and media reports, which are partially self-promotional and may not be comparable across schools. Since the sample is a convenience sample based on what schools had data to collect, this could also introduce more bias into which schools are included. Since it is not clear where all data come from, this is hard to judge. It would be helpful to provide a better description of exact data sources as an appendix.

Reviewer #7: The authors have tried to improve the paper but my main concerns have not been addressed and some of my other comments haven’t been addressed too or haven’t been addressed properly. Therefore I recommend a rejection of the paper.

Comments

1. General comment: I still have significant doubts about the value added of the paper. After reading it I don’t understand what that value added of the paper is.

2. General comment: the paper basically should be in large parts rewritten to make a good and interesting reading.

3. The Introduction is still very superficial. It lacks an indication what data and method will be used in the paper, what value added the paper has and what its structure is.

4. The Literature Review is superficial too. It is difficult to understand and simply boring.

5. The sample description of on page 14 is still very superficial and the reasons for choosing 40 cases of outstanding technician colleges remains still unclear.

6. A similar comment concerns the 16 technician colleges selected as the final research objects.

Reviewer #8: There are no additional comments. All comments made in the first round have been adequately addressed.

7. PLOS authors have the option to publish the peer review history of their article (what does this mean? ). If published, this will include your full peer review and any attached files.

**Do you want your identity to be public for this peer review?** For information about this choice, including consent withdrawal, please see our Privacy Policy .

Reviewer #1: No

Reviewer #2: No

Reviewer #3: No

Reviewer #4: No

Reviewer #5: No

Reviewer #7: No

Reviewer #8: No

---

## [Author Response · Author response to Decision Letter 2]

28 Jul 2025

We appreciate the time and effort that you and the reviewers dedicated to providing feedback on our manuseript and are grateful for the insightfulcomments on and valuable improvements to our paper.

---

## [Decision Letter · Decision Letter 2]

22 Aug 2025

PONE-D-25-07768R2How to Promote the Quality of Technical Education�A Fuzzy-set Qualitative Analysis of 16 Technical Colleges in ChinaPLOS ONE

Dear Dr. Zhang,

Thank you for submitting your manuscript to PLOS ONE. After careful consideration, we feel that it has merit but does not fully meet PLOS ONE’s publication criteria as it currently stands. Therefore, we invite you to submit a revised version of the manuscript that addresses the points raised during the review process.

We look forward to receiving your revised manuscript.

Kind regards,

Pengpeng Ye

Academic Editor

PLOS ONE

Journal Requirements:

Reviewers' comments:

Reviewer's Responses to Questions

**Comments to the Author**

1. If the authors have adequately addressed your comments raised in a previous round of review and you feel that this manuscript is now acceptable for publication, you may indicate that here to bypass the “Comments to the Author” section, enter your conflict of interest statement in the “Confidential to Editor” section, and submit your "Accept" recommendation.

Reviewer #7: (No Response)

2. Is the manuscript technically sound, and do the data support the conclusions?

Reviewer #7: Partly

3. Has the statistical analysis been performed appropriately and rigorously? 

Reviewer #7: Yes

4. Have the authors made all data underlying the findings in their manuscript fully available?

Reviewer #7: No

5. Is the manuscript presented in an intelligible fashion and written in standard English?

Reviewer #7: Yes

6. Review Comments to the Author

Reviewer #7: The authors have tried to improve the paper further and have addressed some of my main concerns but not all. Therefore I recommend a major revision of the paper.

Comments

1. The Introduction is still very superficial and besides one new sentence nothing has changed in the Introduction. It still lacks an indication what data and method will be used in the paper, what value added the paper has and what its structure is. I totally don’t understand why the authors mention those issues in their reply to my comment e.g. “The value of this study is reflected in three dimensions: At the theoretical level, it breaks through the limitations of traditional single-factor analysis, reveals the "social service-driven" mechanism for the high-quality development of TVET from a configurational perspective, and enriches the theoretical framework for improving the quality of vocational education; At the practical level, it identifies key pathways such as "participating in government-funded vocational training programs" and provides operable empirical evidence for institutions to optimize their development strategies; At the methodological level, it verifies the applicability of fsQCA in the field of vocational education and offers a transferable methodological reference for similar global studies (especially meaningful for the configurational analysis of TVET institutions in developing countries). 3. The structure of this paper is arranged as follows: The second part sorts out the relevant theories and existing studies on TVET quality improvement, and clarifies the research gaps; The third part elaborates on the research design, including case selection criteria, data sources, variable measurement, and the fsQCA analysis process; The fourth part presents the results of the fsQCA analysis and explains the key configurations and bottleneck types of high-quality development; The fifth part discusses the theoretical connotations and practical implications of the research findings, responding to the practical needs of global TVET development; The sixth part summarizes the research limitations and future research directions” but fail to include them in the Introduction.

7. PLOS authors have the option to publish the peer review history of their article (what does this mean? ). If published, this will include your full peer review and any attached files.

**Do you want your identity to be public for this peer review?** For information about this choice, including consent withdrawal, please see our Privacy Policy .

Reviewer #7: No

---

## [Author Response · Author response to Decision Letter 3]

25 Aug 2025

Thank you for giving us the opportunity to submit a revised draft of the manuscript “How to Promote the Quality of Technical Education�A Fuzzy-set Qualitative Analysis of 16 Technical Colleges in China” for publication in the Journal of PLOS ONE. We appreciate the time and effort that you and the reviewers dedicated to providing feedback on our manuscript and are grateful for the insightful comments on and valuable improvements to our paper.

First, we would like to specifically acknowledge the meticulous and professional feedback on the Introduction section, which has provided us with critical insights to address its previously identified limitations. Your pointed observations helped us clearly recognize gaps in the original draft, including the lack of explicit articulation of research data sources, methodological rationale, the theoretical and practical value of the study, and the manuscript’s structural roadmap. Guided by your comments, we have refined the Introduction to enhance its depth, coherence, and academic rigor—not only to address these gaps but also to better contextualize the study within existing literature and highlight its unique contributions. This revision process has not only improved the manuscript itself but also deepened our understanding of how to construct a logically rigorous and compelling academic introduction, for which we are truly thankful.

We have carefully checked the revised Introduction to ensure that all the key information you noted (data, methods, study value, structure) is fully and organically integrated, rather than superficially added. This revision not only makes the Introduction more comprehensive and rigorous but also helps readers quickly grasp the core logic and contributions of the study—directly addressing the “superficiality” issue you identified.

Once again, we would like to thank you for your patient guidance and valuable insights. Your comments have been pivotal to improving the rigor and clarity of our manuscript. We have attached the revised manuscript for your review and welcome any further suggestions you may have. We remain committed to making additional refinements to meet the high standards of PLOS ONE.

---

## [Editor Report · Decision Letter 3]

27 Aug 2025

How to Promote the Quality of Technical Education�A Fuzzy-set Qualitative Analysis of 16 Technical Colleges in China

PONE-D-25-07768R3

Dear Dr. Zhang,

We’re pleased to inform you that your manuscript has been judged scientifically suitable for publication and will be formally accepted for publication once it meets all outstanding technical requirements.

Kind regards,

Pengpeng Ye

Academic Editor

PLOS ONE
---

## [Editor Report · Acceptance letter]

PONE-D-25-07768R3

PLOS ONE

Dear Dr. Zhang,

I'm pleased to inform you that your manuscript has been deemed suitable for publication in PLOS ONE. Congratulations! Your manuscript is now being handed over to our production team.

Kind regards,

on behalf of

Dr. Pengpeng Ye

Academic Editor

PLOS ONE